

# From Abstainers to Dependent Drinkers: alcohol consumption patterns and risk factors among Portuguese university students

Lucas Saldanha[1], Alberto Crego[1], Natália Almeida-Antunes[2],
Rui Rodrigues[1], Adriana Sampaio[1] and Eduardo López-Caneda[1]

[1] Psychological Neuroscience Laboratory, Psychology Research Center (CIPsi), School of Psychology, University of Minho, Braga, Portugal
[2] RISE-Health, Center for Translational Health and Medical Biotechnology Research (TBIO), ESS, Polytechnic of Porto, Porto, Portugal

Corresponding author
Eduardo López-Caneda,
eduardo.lopez@psi.uminho.pt

## ABSTRACT

**Background:** Alcohol is the most prevalent psychoactive substance consumed around the world. In Portugal, alcohol consumption is deeply embedded in social and cultural practices, contributing to high prevalence rates among university students, with binge drinking emerging as a predominant consumption pattern. Despite the associations between this drinking behaviour and numerous social, physical, and psychological problems, research on alcohol consumption in Portuguese university populations remains limited. Thus, this study aimed to provide a comprehensive description of alcohol use patterns among a large sample of Portuguese university students, focusing on adolescents and young adults.

**Methods:** A total of 1,746 students, aged 17–24 years, participated in a cross-sectional study and were surveyed using the Alcohol Use Disorders Identification Test (AUDIT) and additional questionnaires regarding socio-demographic information, alcohol and illicit drug use, smoking habits, and alcohol cravings. Statistical analyses included descriptive statistics, group comparisons, and multinomial logistic regressions to obtain odds ratios (ORs) for group membership.

**Results:** Alcohol consumption was reported by 83.2% of students over the past year. Based on their drinking patterns and AUDIT score, participants were distributed across five drinking groups: Abstainers (16.8%), Moderate Drinkers (35.1%), Hazardous Drinkers (25.8%), Binge Drinkers (20.8%), and Dependent Drinkers (1.5%). Nearly 47% of students revealed harmful drinking patterns, and 1.5% exhibited symptoms of alcohol dependence. A progressive increase in the severity of alcohol consumption characteristics was observed across the groups, with Dependent Drinkers reporting the highest levels of all assessed characteristics. Significant predictors of group membership included polydrug use, standard weekly consumption, earlier drinking onset, and higher levels of alcohol craving. Polydrug use, reported by 27.3% of students, was the strongest predictor for being a Hazardous Drinker (OR = 10.75), Binge Drinker (OR = 13.20), and Dependent Drinker (OR = 21.40). Binge Drinkers displayed standard weekly consumption and craving levels comparable to Dependent Drinkers, while Moderate Drinkers exhibited the

least risky patterns among drinkers, including a later age of onset of drinking. Male students reported significantly greater consumption and craving levels than their female peers.

**Conclusions:** This study highlights the prevalence of harmful drinking behaviours among Portuguese university students and identifies critical risk factors, such as polydrug use and early drinking onset. These findings underscore the need for prevention programmes focused on delaying the onset of alcohol use, reducing drug use, and promoting healthier behaviours within academic settings.

# INTRODUCTION

Alcohol is the most widely consumed psychoactive substance worldwide (*World Health Organization, 2024*). In 2019, an estimated 44% of the global population above age 15 consumed alcohol, with Europe exhibiting the highest levels of drinking (62%; *World Health Organization, 2024*). Alcohol consumption is a major public health challenge, closely linked to widespread disease and injury (*Rehm et al., 2009*, *2017*). Globally, alcohol is responsible for approximately 2.6 million deaths per year, accounting for 4.7% of all fatalities (*World Health Organization, 2024*). In Portugal, the prevalence of alcohol consumption over the last year was estimated at 62%, and alcohol-related mortality accounted for 2% of all deaths (*Intervention Service for Addictive Behaviors and Dependencies, 2023*).

Similarly, the levels of alcohol use among adolescents and young adults have shown concerning trends in recent years (*World Health Organization, 2024*). The European School Survey Project on Alcohol and Other Drugs reported that an average of 79% of students consume alcohol, with many starting as early as age 13 or younger (*European School Survey Project on Alcohol and Other Drugs, 2020*). High-risk consumption patterns are increasing among youth (*World Health Organization, 2024*), with reports indicating a rise in both daily consumption and intoxication episodes (*Addolorato et al., 2018*; *Balsa, Vital & Urbano, 2023*; *European School Survey Project on Alcohol and Other Drugs, 2020*). Among Portuguese youth aged 15 to 24, the prevalence of alcohol use was 59.2% (*Intervention Service for Addictive Behaviors and Dependencies, 2023*), with 9% of 18-year-olds reporting daily drinking and over 60% experiencing multiple intoxication episodes in the past year (*Carapinha, Calado & Neto, 2024*). Furthermore, nearly 30% of young adults acknowledged they had experienced alcohol-related problems, highlighting the importance of targeted preventive efforts (*Carapinha, Calado & Neto, 2024*).

An increasing number of studies have focused on characterising alcohol consumption patterns among young people, particularly university students (*Gómez et al., 2017*; *Gonçalves & Carvalho, 2017*; *Moutinho, de Oliveira Cruz Mendes & Lopes, 2018*; *Ferreira Alves, Precioso & Becoña, 2021*; *Herrero-Montes et al., 2022*; *Ay et al., 2025*). Over time, alcohol consumption has been normalised as a pleasurable and significant aspect of social

gatherings and festivities (*Hamilton et al., 2020*; *Babor et al., 2022*). However, this cultural normalisation is especially evident in the academic environment, where drinking is often accepted, encouraged, and even admired (*Santos, Veiga & Pereira, 2009*; *Carter, Brandon & Goldman, 2010*; *Cunha, 2014*; *Moutinho, de Oliveira Cruz Mendes & Lopes, 2018*). In addition, research indicates that university students tend to overestimate both the frequency and quantity of their peers' alcohol consumption, as well as the extent to which excessive drinking is socially approved (*Borsari & Carey, 2001*). These misperceptions can, in turn, prompt students to increase their own alcohol use to conform to these perceived norms (*Crawford & Novak, 2007*; *França, Dautzenberg & Reynaud, 2010*). As such, this demographic exhibits high rates of alcohol use, with binge drinking emerging as one of the predominant consumption patterns (*Kuntsche et al., 2017*; *López-Caneda, Cadaveira & Campanella, 2019*).

Binge drinking, characterised by an episodic pattern of excessive drinking followed by periods of low or no consumption, is defined as the consumption of at least five standard drinks for males or four standard drinks for females within 2 h, resulting in a blood alcohol concentration (BAC) of 0.08 g/dL or above (*National Institute on Alcohol Abuse and Alcoholism, 2024b*). This behaviour has become a regular practice in European countries, with 34% of students aged 15–16 reporting a binge drinking episode in the previous month (*European School Survey Project on Alcohol and Other Drugs, 2020*). In Portugal, the prevalence of binge drinking is estimated at 10.3% for the general population, rising to 22.4% among current drinkers aged 15–24 (*Intervention Service for Addictive Behaviors and Dependencies, 2023*). Some studies with Portuguese student samples report even higher rates, such as 50.6% (*Moutinho, de Oliveira Cruz Mendes & Lopes, 2018*), 43.1% (*Santos, Veiga & Pereira, 2009*), and 37% (*Alcântara da Silva et al., 2015*). However, methodological differences, such as the number of drinks for classification of a binge drinking episode and target age ranges, may account for the variation in reported prevalence rates.

On the other hand, it is important to note that binge drinking is not the sole pattern of alcohol use in young adulthood. Other patterns, including moderate drinking, hazardous drinking, and alcohol dependence, are also present at this age (*National Institute on Alcohol Abuse and Alcoholism, 2024a*). Moderate drinking refers to regular but not excessive drinking that does not pose negative social or health consequences (*National Institute on Alcohol Abuse and Alcoholism, 2024a*), while hazardous drinking is associated with an increased risk of harm to oneself or others due to risky drinking practices (*World Health Organization, 2024*). In Portugal, 37.2% of individuals aged 15–24 are classified as moderate drinkers, while fewer than 10% exhibit patterns of hazardous drinking (*Balsa, Vital & Urbano, 2023*). Alcohol dependence, a condition recognised within alcohol use disorder (AUD), is characterised as the inability to control or cease alcohol consumption, despite the harmful effects (*Carvalho et al., 2019*). While the precise diagnosis for AUD requires a comprehensive clinical assessment, estimates indicate that 1.1% of the general Portuguese population experiences dependence symptoms, with prevalence declining to 0.3% among those aged 15–24 (*Intervention Service for Addictive Behaviors and Dependencies, 2023*).
Excessive alcohol consumption patterns are recognised as a major public health concern linked to several social and health consequences (*GBD 2016 Alcohol and Drug Use Collaborators, 2018*). It is a significant contributor to a range of diseases, including cancer, cardiovascular conditions, diabetes, and mental health disorders (*Wilsnack et al., 2018*; *GBD 2016 Alcohol Collaborators, 2018*; *Perez-Araluce et al., 2023*). Social consequences include a variety of issues such as vandalism, physical aggression, work absenteeism and accidents, legal troubles, and family conflicts, as well as more severe outcomes like child maltreatment, increased vulnerability to sexual assault, and involvement in driving accidents (*Karriker-Jaffe et al., 2018*; *Nayak et al., 2019*; *Babor et al., 2022*). Among adolescents and young people, these consequences are primarily manifested in poor academic performance, interpersonal violence, drunk driving, and risky sexual behaviours (*Miller et al., 2007*; *Calafat et al., 2011*; *Kuntsche et al., 2017*; *Wilsnack et al., 2018*; *Lukács et al., 2021*; *Pentz et al., 2023*). This developmental stage, spanning adolescence to early adulthood, is marked by critical brain maturation, particularly in the prefrontal cortex, responsible for high-level cognitive processes, including decision-making and inhibitory control (*Jones, Lueras & Nagel, 2018*). Heavy alcohol consumption during this period can disrupt normative brain developmental, accentuating gray and white matter disruptions, and is associated with impairments in attention, memory, and executive functions, which can persist into adulthood (*Bava & Tapert, 2010*; *López-Caneda et al., 2014*; *Lees et al., 2019*; *Tapert & Eberson-Shumate, 2022*). Consequently, adolescence and young adulthood represent highly vulnerable periods to the neurotoxic effects of alcohol on the brain (*Crews, He & Hodge, 2007*; *Østby et al., 2009*; *Crews et al., 2016*). Likewise, an increasing body of literature highlights the link between binge drinking during these periods and both structural and functional brain anomalies, as well as several cognitive deficits (*Carbia et al., 2018*; *Lees et al., 2019*; *Almeida-Antunes et al., 2021*). Furthermore, due to the impact of these patterns on psychological well-being, excessive alcohol use among university students has been linked to lower mental quality of life (*Perez-Araluce et al., 2023*), higher rates of depression (*Kenney et al., 2018*), and increased suicide attempts (*Miller et al., 2007*).

While previous studies have examined alcohol consumption among young Portuguese individuals, most have targeted broad age groups, such as 18–54 years (*e.g.*, *Ferreira Alves, Precioso & Becoña, 2021*) or 18–63 years (*e.g.*, *Moreira et al., 2020*), and none have specifically targeted adolescents and young university students–a critical transitional period into early adulthood. During this stage, drinking behaviours are often established, particularly within university settings, where social interactions and cultural norms frequently promote alcohol consumption, potentially leading to lasting implications for both health and academic performance (*Borsari, Murphy & Barnett, 2007*). Additionally, many of these studies have been constrained by small sample sizes (*e.g.*, *Ferreira, Moutinho & Teixeira, 2019* ($n = 131$); *Pentz et al., 2023* ($n = 204$); *Valentim, Moutinho & Carvalho, 2021* ($n = 171$)) and/or have primarily focused on changes in alcohol consumption patterns resulting from the COVID-19 pandemic and its associated restrictions (*Vasconcelos et al., 2021*; *Oliveira et al., 2023*), which may impact the generalisability of their findings.

Thus, the present study, which includes a sample of nearly 1,800 participants, aims to provide a more accurate and comprehensive reflection of alcohol consumption patterns among Portuguese university students, specifically targeting the adolescent and young adult age group. In addition to offering a robust epidemiological update, this study examines key behavioural and psychological predictors–namely, polydrug use, craving levels, and early onset of drinking–to identify risk profiles associated with harmful alcohol use in vulnerable youth.

## METHODS

Portions of this text were previously published as part of a preprint (*Saldanha et al., 2025*).

### Participants

A total of 1,825 students from the University of Minho (Braga, Portugal) were recruited. Participants who did not meet age or grouping criteria were excluded from the study. The final sample comprised 1,746 students (67.6% female, 31.6% male, and 0.8% of unspecified sex), aged between 17 and 24 years (M = 19.62, SD = 1.39).

Participants were recruited through a screening questionnaire, which included the Alcohol Use Disorder Identification Test (AUDIT; *Babor et al., 2001*), along with other questions concerning alcohol and drug use. Five groups were formed based on their drinking patterns and AUDIT scores. Students who reported a total AUDIT score equal to 0 were classified as Abstainers, noting that, as such, some participants in this group may have previously consumed alcohol or do so on sporadic occasions. Participants with an AUDIT score of ≤4 for men or ≤3 for women were classified as Moderate Drinkers. Those who reported an AUDIT score between 5 and 19 for men or between 4 and 19 for women were classified as Hazardous Drinkers. Students who reported (i) Hazardous Drinker criteria, (ii) drinking 5 or more drinks on one occasion at least once a month (*i.e.*, rated ≥2 on AUDIT question 3), and (iii) drinking at a speed of at least two drinks per hour during these episodes were classified as Binge Drinkers (*National Institute on Alcohol Abuse and Alcoholism, 2024b*). Finally, students with an AUDIT score of ≥20 were classified as Dependent Drinkers–reflecting the presence of problems and typical symptoms consistent with a probable diagnosis of dependence according to AUDIT criteria (*Babor et al., 2001*). The groups were constructed according to the AUDIT score groupings in *Balsa, Vital & Urbano (2023)*, used in the National Inquiry for Substance Consumption in the General Population in Portugal.

A total of 68 students were excluded for not meeting the age criteria (*i.e.*, age > 24 years). Additionally, seven students were excluded for classifying as false Binge Drinkers (*i.e.*, AUDIT score of ≤4 for men and ≤3 for women). Finally, four students were excluded due to missing data, which prevented their classification into any of the groups.

### Measures

Questionnaires regarding socio-demographic data, pattern and amount of alcohol consumption, smoking habits, use of illicit drugs, and alcohol cravings were administered. For the characterisation of alcohol consumption, the following data were collected: type

and amount of drinks per week, speed of consumption, age of onset of drinking, and percentage of times participants got drunk while drinking. Data concerning other types of consumption included smoking frequency, as well as type and frequency of illicit drug use, and polydrug use, defined as the combined use of two or more different substances. The frequency of use was evaluated using a 4-point Likert scale.

The Portuguese version of the AUDIT (*Cunha, 2002*) was used to assess alcohol consumption patterns. This 10-item self-report questionnaire evaluates alcohol use, dependence symptoms, and alcohol-related problems, including risky and harmful drinking, over the past 12 months. The levels of alcohol cravings were assessed through the Portuguese versions of the Penn Alcohol Craving Scale (PACS; *Flannery, Volpicelli & Pettinati, 1999*; *Pombo, Ismail & Cardoso, 2008*) and the Alcohol Craving Questionnaire–Short Form Revised (ACQ-SF-R; *Singleton, 1995*; *Rodrigues et al., 2021*). The PACS is a 5-item self-report questionnaire that assesses the frequency, intensity, and duration of alcohol-related thoughts, the capacity for alcohol restraint, and overall craving levels over the past week. The ACQ-SF-R is a 12-item self-report questionnaire that measures acute alcohol craving across three dimensions: *Emotionality*, related to the relief from withdrawal or negative affect; *Purposefulness*, related to the planning and intent to drink; and *Compulsivity*, related to the loss of control over drinking. On both scales, higher scores indicate stronger cravings, with a score of ≥20 on the PACS signifying clinical significance. The authors have permission to use these instruments from the copyright holders.

A cross-sectional study was performed. The study was conducted in accordance with the Declaration of Helsinki, meeting the requirements for exemption and anonymity, and received approval by the Institutional Ethics Committee for Social Sciences and Humanities of the University of Minho (UM), Braga, Portugal (CECSH 078/2018). During the academic years of 2018–2019 and 2019–2020, *i.e.*, before the COVID-19 pandemic, a set of the above-listed questionnaires was administered in the classrooms of several UM courses, using a pen-and-paper format, following participants' verbal consent.

## Statistical analysis

Statistical analysis was performed using the software IBM SPSS Statistics, version 29 (IBM Corp., Armonk, NY, USA). To determine whether participants were consumers or non-consumers, the original frequency variables for consumption of alcohol and illicit drugs were recoded into binary variables. The dichotomous variables were used in the subsequent analysis to examine the prevalence of substance use. An exploratory analysis, including normality tests, was initially conducted, revealing non-normal distributions overall. Descriptive statistics were calculated to summarise the sample characteristics, incorporating the central tendency and dispersion measures for continuous variables and the absolute and relative frequencies for categorical variables. Non-parametric tests were used to explore differences between variables, and Pearson's Chi-square tests were applied to assess associations between categorical variables and the drinking groups. To explore associations between the continuous variables and the drinking groups, a Welch's analysis

of variance (ANOVA) was conducted, with Games-Howell *post-hoc* tests for paired comparisons, with an alpha level <0.05. Lastly, multinominal logistic regressions were conducted to obtain the odds ratios (ORs) for the independent variables from the final group's model. The model included the following variables: sex, school, drinks in a standard week, age of onset of drinking, percentage of drunkenness, and polydrug use. In an additional model, we also included PACS and ACQ-SF-R total scores variables. The models were tested for multicollinearity, and confidence intervals of 95% were reported.

Since data collection was conducted through a pen-and-paper format, some participants did not respond to all of the questions, resulting in missing data for specific variables. Missing data imputation was carried out for group allocation variables using the medians of the AUDIT question 3 score and speed of consumption ($n = 4$). Sex was not inputted; however, participants missing this information were included if it did not impact group allocation ($n = 14$). For the variables where imputation was unfeasible, data were left incomplete, reducing the sample sizes for some analyses. As the data appears to be missing at random, it is unlikely to significantly bias the results (*Osborne, 2013*). Since age was an exclusion factor, the proportion of participants removed based on age (>24 years) was calculated (3.7% of the total sample). This percentage was used to estimate a potential error among all participants with missing age data (16.3%), resulting in an approximate error margin of 0.72% for age misclassification within the total sample.

## RESULTS

In total, 1,746 students were included in the analysis. The overall sample consisted of 67.6% females, 31.6% males, and 0.08% of unspecified sex. Most of the students who participated in the study were from social sciences courses (49.4%), followed by health sciences (21.9%), engineering and technological sciences (15.5%), arts and humanities (3.3%), and natural sciences (3.0%) (see Fig. S1 in the Supplemental Material). Alcohol consumption over the last 12 months was reported by 83.2% of students, with 86.8% of males and 81.4% of females being consumers. The sample consisted of 294 Abstainers (16.8%), 612 Moderate Drinkers (35.1%), 451 Hazardous Drinkers (25.8%), 363 Binge Drinkers (20.8%), and 26 Dependent Drinkers (1.5%). Summarised data for each group is available in Table 1.

### Alcohol consumption characteristics
#### *Drinking characteristics*
Among students with alcohol consumption over the last year, the mean AUDIT total score was 5.81 (SD = 4.90), the mean age of drinking onset was 16.44 (SD = 1.50), the mean percentage of alcohol consumption resulting in drunkenness was 28.04% (SD = 31.74), and the mean number of drinks consumed in a standard week was 5.77 (SD = 7.22) (Table 1).

**AUDIT total scores.** Significant differences in AUDIT total scores were observed between drinking groups [$\chi^2(3) = 1,153.40$, $p < 0.001$] (Fig. 1A). *Post-hoc* Games-Howell pairwise comparisons revealed significant differences for all pairwise comparisons ($p < 0.001$), with the highest AUDIT scores reported for Dependent Drinkers.

**Table 1 Demographics and consumption characteristics.**

| | Total sample | | | Drinking groups | | | | | | | | | | | | | | |
|---|---|---|---|---|---|---|---|---|---|---|---|---|---|---|---|---|---|---|
| | | | | Abstainers | | | Moderate drinkers | | | Hazardous drinkers | | | Binge drinkers | | | Dependent drinkers | | |
| | Male | Female | Total[a] | Male | Female | Total[b] | Male | Female | Total[c] | Male | Female | Total[d] | Male | Female | Total[e] | Male | Female | Total[f] |
| **Demographic measures** | | | | | | | | | | | | | | | | | | |
| $n$ | 552 | 1,180 | 1,746 | 73 | 220 | 294 | 117 | 430 | 612 | 119 | 328 | 451 | 171 | 190 | 363 | 12 | 12 | 26 |
| Age | 19.65 ± 1.60 | 19.61 ± 1.28 | 19.62 ± 1.39 | 19.38 ± 1.50 | 19.65 ± 1.30 | 19.58 ± 1.35 | 19.69 ± 1.57 | 19.65 ± 1.29 | 19.66 ± 1.37 | 20.10 ± 1.74 | 19.53 ± 1.31 | 19.68 ± 1.45 | 19.38 ± 1.59 | 19.62 ± 1.21 | 19.52 ± 1.39 | 19.78 ± 0.67 | 19.56 ± 1.13 | 19.67 ± 0.91 |
| **Drinking characteristics** | | | | | | | | | | | | | | | | | | |
| Total AUDIT score | 6.19 ± 5.53 | 4.16 ± 4.47 | 4.83 ± 4.97 | 0 ± 0.0 | 0 ± 0.0 | 0 ± 0.0 | 2.45 ± 1.15 | 1.70 ± 0.81 | 1.92 ± 0.98 | 7.65 ± 2.90 | 6.12 ± 2.55 | 6.56 ± 2.74 | 10.30 ± 3.58 | 9.88 ± 3.47 | 10.06 ± 3.53 | 26.08 ± 4.72 | 24.17 ± 3.97 | 25.27 ± 4.46 |
| Age of onset of drinking | 16.22 ± 1.58 | 16.59 ± 1.43 | 16.47 ± 1.49 | 17.25 ± 1.17 | 16.92 ± 1.14 | 16.97 ± 1.13 | 16.75 ± 1.55 | 16.89 ± 1.52 | 16.85 ± 1.53 | 16.15 ± 1.51 | 16.44 ± 1.25 | 16.36 ± 1.33 | 15.85 ± 1.44 | 16.18 ± 1.42 | 16.03 ± 1.43 | 14.18 ± 1.60 | 15.33 ± 1.16 | 14.88 ± 1.48 |
| Drinks in a standard week | 6.63 ± 8.20 | 3.93 ± 6.09 | 4.80 ± 6.93 | – | – | – | 2.55 ± 3.31 | 1.56 ± 2.41 | 1.85 ± 2.73 | 7.53 ± 6.63 | 5.20 ± 5.33 | 5.81 ± 5.78 | 12.73 ± 9.12 | 10.74 ± 8.31 | 11.67 ± 8.73 | 14.08 ± 15.71 | 20.25 ± 8.95 | 16.73 ± 12.49 |
| Percentage of drunkenness | 26.31 ± 30.88 | 21.86 ± 30.56 | 23.37 ± 30.78 | – | – | – | 14.02 ± 25.10 | 7.07 ± 17.51 | 9.01 ± 20.13 | 29.62 ± 28.21 | 35.15 ± 30.81 | 33.91 ± 30.26 | 44.47 ± 29.72 | 54.92 ± 29.95 | 49.89 ± 30.31 | 79.82 ± 16.50 | 68.75 ± 18.96 | 74.12 ± 17.60 |
| **Five drinks or more in a single occasion[*] (%)** | | | | | | | | | | | | | | | | | | |
| Never | 31.9% | 48.7% | 43.4% | 100.0% | 100.0% | 100.0% | 56.5% | 77.9% | 71.7% | 2.5% | 6.1% | 5.3% | – | – | – | – | – | – |
| Less than once a month | 28.3% | 29.2% | 28.8% | – | – | – | 42.4% | 22.1% | 27.9% | 67.2% | 76.3% | 73.4% | – | – | – | 8.3% | – | 3.8% |
| At least once a month | 26.3% | 16.9% | 19.9% | – | – | – | 1.1% | – | 0.3% | 21.8% | 14.3% | 16.6% | 68.4% | 78.9% | 74.1% | – | 16.7% | 7.7% |
| At least once a week | 12.7% | 4.7% | 7.2% | – | – | – | – | – | – | 8.4% | 3.0% | 4.4% | 31.0% | 19.5% | 24.8% | 58.3% | 66.7% | 61.5% |
| Daily or almost daily | 0.9% | 0.5% | 0.7% | – | – | – | – | – | – | – | 0.3% | 0.2% | 0.6% | 1.6% | 1.1% | 33.3% | 16.7% | 26.9% |
| **Alcohol craving measures** | | | | | | | | | | | | | | | | | | |
| Total PACS score | 2.51 ± 3.34 | 1.77 ± 3.31 | 2.01 ± 3.33 | 0 ± 0.0 | 0.08 ± 0.36 | 0.06 ± 0.31 | 1.25 ± 1.85 | 0.56 ± 1.24 | 0.77 ± 1.48 | 2.47 ± 2.80 | 2.63 ± 3.39 | 2.59 ± 3.23 | 3.93 ± 3.26 | 4.07 ± 3.78 | 4.01 ± 3.53 | 9.40 ± 5.46 | 16.40 ± 7.16 | 11.73 ± 6.75 |
| Total ACQ-SF-R score | 2.17 ± 0.99 | 1.86 ± 0.87 | 1.99 ± 0.93 | 1.34 ± 0.61 | 1.20 ± 0.39 | 1.25 ± 0.48 | 1.89 ± 0.73 | 1.55 ± 0.60 | 1.68 ± 0.67 | 2.27 ± 0.84 | 2.21 ± 0.84 | 2.23 ± 0.83 | 2.67 ± 1.00 | 2.52 ± 0.85 | 2.6 ± 0.94 | 3.53 ± 1.28 | 3.47 ± 1.91 | 3.51 ± 1.45 |
| Emotionality factor | 2.10 ± 1.22 | 1.70 ± 1.04 | 1.86 ± 1.13 | 1.33 ± 0.78 | 1.15 ± 0.42 | 1.21 ± 0.58 | 1.93 ± 1.10 | 1.47 ± 0.81 | 1.65 ± 0.96 | 2.26 ± 1.18 | 1.95 ± 1.06 | 2.04 ± 1.10 | 2.42 ± 1.32 | 2.19 ± 1.19 | 2.32 ± 1.27 | 3.20 ± 1.31 | 3.96 ± 2.87 | 3.45 ± 1.89 |
| Purposefulness factor | 2.92 ± 1.55 | 2.63 ± 1.61 | 2.74 ± 1.59 | 1.67 ± 1.46 | 1.48 ± 1.19 | 1.55 ± 1.29 | 2.50 ± 1.27 | 2.07 ± 1.27 | 2.24 ± 1.28 | 2.99 ± 1.33 | 3.26 ± 1.42 | 3.19 ± 1.40 | 3.79 ± 1.38 | 3.88 ± 1.59 | 3.83 ± 1.48 | 4.17 ± 1.83 | 3.87 ± 2.21 | 4.07 ± 1.89 |
| Compulsivity factor | 1.70 ± 0.98 | 1.47 ± 0.76 | 1.56 ± 0.86 | 1.07 ± 0.27 | 1.04 ± 0.18 | 1.05 ± 0.21 | 1.37 ± 0.47 | 1.25 ± 0.44 | 1.29 ± 0.46 | 1.71 ± 0.85 | 1.73 ± 0.86 | 1.72 ± 0.85 | 2.15 ± 1.18 | 1.91 ± 0.90 | 2.04 ± 1.07 | 3.45 ± 1.51 | 2.55 ± 2.22 | 3.15 ± 1.75 |
| **Substance use (%)** | | | | | | | | | | | | | | | | | | |
| Tobacco smokers | 25.9% | 21.2% | 22.7% | 2.7% | 1.8% | 2.0% | 8.5% | 12.1% | 11.1% | 31.9% | 31.1% | 31.5% | 45.6% | 43.7% | 44.6% | 83.3% | 33.3% | 73.1% |
| Cannabis users | 18.1% | 8.5% | 11.5% | – | 0.5% | 0.3% | 7.3% | 3.3% | 4.4% | 18.5% | 13.1% | 14.6% | 34.5% | 20.5% | 27.0% | 50.0% | 25.0% | 34.6% |
| Hallucinogens users | 2.2% | 1.2% | 1.5% | – | – | – | 1.7% | 0.2% | 0.7% | 2.5% | 1.8% | 2.0% | 2.9% | 2.9% | 3.3% | 8.3% | – | 3.8% |
| Amphetamines users | 0.5% | 0.5% | 0.5% | – | – | – | – | – | – | 0.8% | – | 0.2% | – | 2.6% | 1.4% | 16.7% | 8.3% | 11.5% |
| Cocaine users | 0.9% | 0.4% | 0.6% | – | – | – | – | – | – | 1.7% | – | 0.4% | 0.6% | 1.6% | 1.1% | 16.7% | 16.7% | 15.4% |
| Ecstasy users | 1.1% | 1.0% | 1.0% | – | – | – | 0.6% | 0.5% | 0.5% | 0.8% | 0.6% | 0.7% | 1.2% | 3.2% | 2.2% | 16.7% | 16.7% | 15.4% |
| Heroin users | 0.5% | 0.2% | 0.3% | – | – | – | – | 0.2% | 0.2% | 0.8% | – | 0.2% | – | – | – | 16.7% | 8.3% | 11.5% |
| Polydrug users | 32.6% | 24.7% | 27.3% | – | 0.9% | 0.7% | 13.6% | 14.7% | 14.4% | 38.7% | 37.5% | 37.9% | 58.5% | 50.0% | 54.0% | 83.3% | 66.7% | 73.1% |

**Notes:**

Unspecified sex: [a] $n = 14$; [b] $n = 1$; [c] $n = 5$; [d] $n = 4$; [e] $n = 2$; [f] $n = 2$.

[*] AUDIT, Question 3–Frequency of consumption of 5 or more alcoholic drinks in a single occasion.

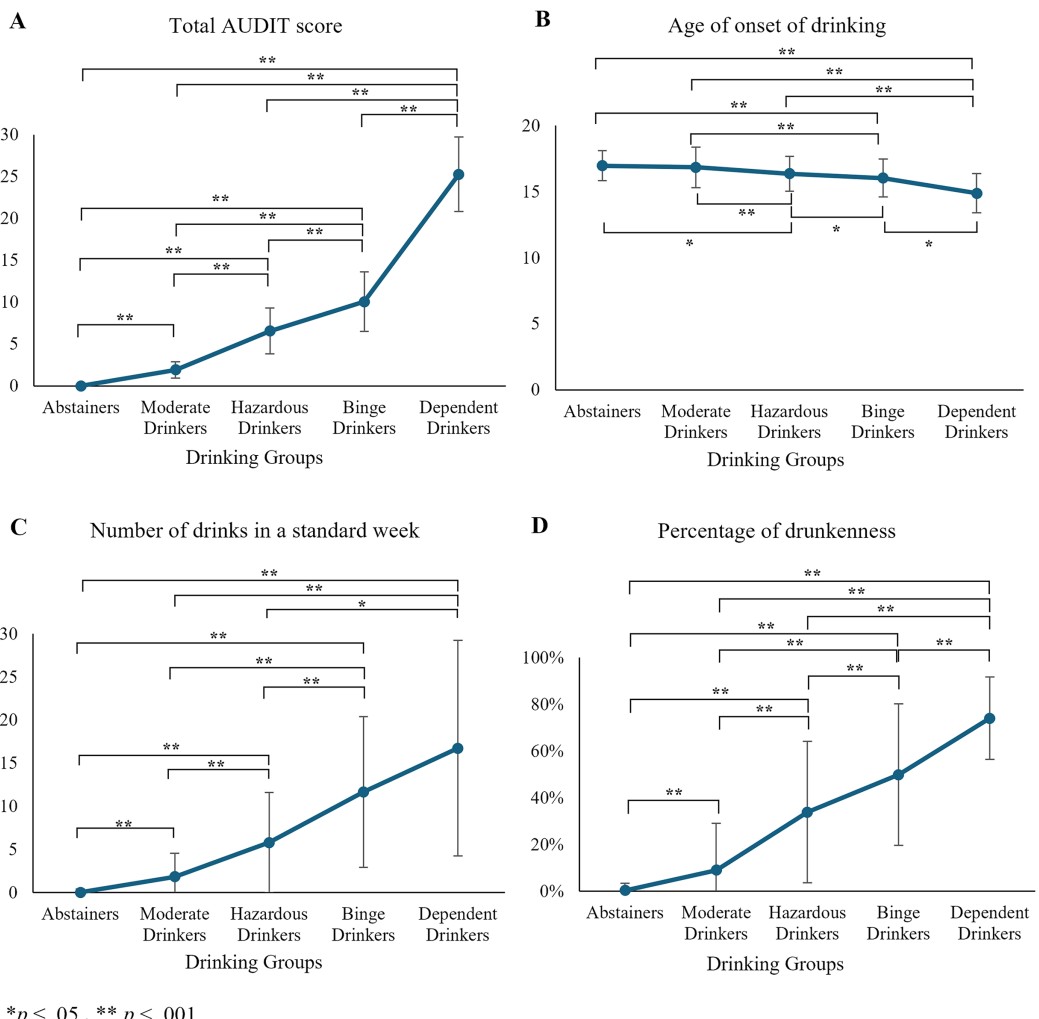

*p ≤ .05 , ** p < .001

**Figure 1** **Mean differences across drinking groups for (A) total AUDIT score, (B) age of onset of drinking, (C) number of drinks in a standard week, and (D) percentage of drunkenness.**

**Age of onset of drinking.** The analysis of variance revealed significant differences between groups in the age of drinking onset [F(4, 135.946) = 27.608, $p < 0.001$, $\omega^2 = 0.071$] (Fig. 1B). *Post-hoc* analysis using the Games-Howell test identified significant differences across all group comparisons, showing a linear progression from Abstainers to Dependent Drinkers, with the latter starting at the youngest ages. Hazardous Drinkers reported earlier ages than Moderate Drinkers ($p < 0.001$) and Abstainers ($p = 0.002$). Binge Drinkers reported earlier ages than Hazardous Drinkers ($p = 0.008$), Moderate Drinkers ($p < 0.001$), and Abstainers ($p < 0.001$). Dependent Drinkers reported earlier ages than Binge Drinkers ($p = 0.007$), Hazardous ($p < 0.001$), Moderate Drinkers ($p < 0.001$), and Abstainers ($p < 0.001$).

**Drinks in a standard week.** Results also showed significant differences in the number of alcoholic drinks consumed in a standard week [$F(4, 171.899) = 336.345$, $p < 0.001$, $\omega^2 = 0.393$] (Fig. 1C), with *post-hoc* analysis revealing significant differences between all

group comparisons, except between Binge Drinkers and Dependent Drinkers, with the latter reporting the highest number of drinks. Specifically, Moderate Drinkers reported a higher number of drinks than Abstainers ($p < 0.001$). Hazardous Drinkers reported more drinks than Moderate Drinkers ($p < 0.001$), and Abstainers ($p < 0.001$). Binge Drinkers reported a higher number of drinks than Hazardous Drinkers ($p < 0.001$), Moderate Drinkers ($p < 0.001$) and Abstainers ($p < 0.001$). Dependent Drinkers reported more drinks than Hazardous Drinkers ($p = 0.001$), Moderate Drinkers ($p < 0.001$), and Abstainers ($p < 0.001$).

**Percentage of drunkenness.** Finally, the analysis also revealed significant differences in the percentage of drunkenness [$F(4, 166.897) = 481.707$, $p < 0.001$, $\omega^2 = 0.392$] (Fig. 1D), with *post-hoc* tests showing significant differences between all group comparisons ($p < 0.001$ for all pair comparisons), with Dependent Drinkers having the highest percentage.

### Drinking patterns

Attending to the overall sample, a large proportion of students reported having previously engaged in the consumption of 5 or more alcoholic drinks on a single occasion (56.6%). Additionally, regarding the students who consumed alcohol, most of them engaged in this type of consumption less than once a month (34.6%). The majority of students reporting monthly consumption of this pattern belong to the Binge Drinkers group (74.1%). Weekly consumption was predominantly observed in the Dependent Drinkers group (61.5%), with the highest rate of daily or almost daily consumption (26.9%) among the groups.

Regarding the speed of consumption, alcohol-drinking students mostly reported consuming one drink in 2 hours (25.1%). Moderate Drinkers reported a higher frequency of consuming one drink over 3 hours or more (38.1%), while Hazardous Drinkers showed higher frequencies for consuming one drink in 2 hours (30.2%). Binge Drinkers reported the highest frequencies for drinking at speeds of 2 or 3 drinks per hour (38.8% and 34.4%, respectively). Finally, Dependent Drinkers showed the highest rates for consuming 4 drinks per hour (19.2%) and 7 or more drinks per hour (19.2%).

### Alcoholic beverages consumption

The most commonly consumed alcoholic beverages among students were beer (36.9%) and shots (36.3%). Although the proportions of alcohol types consumed varied across drinking groups, beer and shots remained the most frequently consumed drinks for all groups (see Fig. S2). Moderate and Hazardous Drinkers displayed similar patterns, with Moderate Drinkers reporting nearly equal consumption of beer (22.1%) and shots (22.9%), while Hazardous Drinkers showed slightly higher consumption of beer (51.7%) than shots (47.5%). In contrast, Binge Drinkers and Dependent Drinkers exhibited high levels of consumption across all types of alcohol, with Binge Drinkers reporting the highest beer consumption (71.1%) and Dependent Drinkers the highest consumption of shots (84.6%).

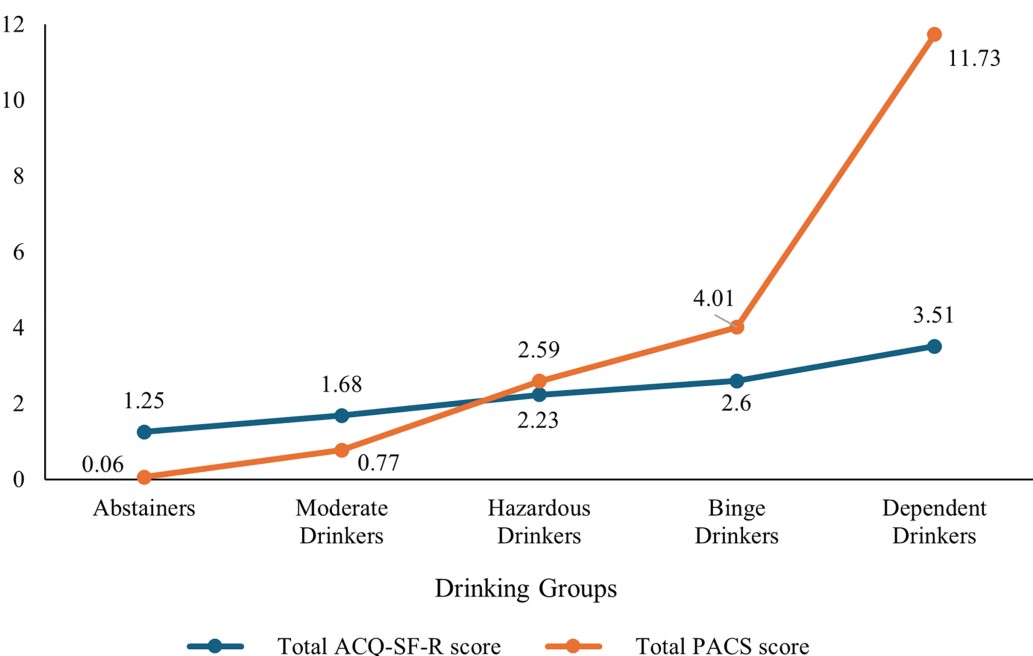

**Figure 2 Mean differences across drinking groups for PACS and ACQ-SF-R total scores.**

### Substance use

Apart from alcohol consumption, 27.3% of students were identified as polydrug users. Tobacco was the most commonly used substance across all groups (22.7%), followed by cannabis (11.5%) (see Fig. S3). Polydrug use was present across all groups, with 41.2% of polydrug users being Binge Drinkers. Within the drinking groups, over half of Binge Drinkers were polydrug users (54.0%), while Dependent Drinkers showed the highest intra-group prevalence of polydrug users (73.1%). The use of other substances–such as hallucinogens, amphetamines, cocaine, ecstasy, and heroin–was relatively low across groups, except among Dependent Drinkers, who presented notably higher rates for amphetamines (11.5%), cocaine (15.4%), ecstasy (15.4%) and heroin (11.5%).

### Craving levels

Craving levels in the total sample were reported with a mean PACS total score of 2.01 (SD = 3.33) and a mean ACQ-SF-R total score of 1.99 (SD = 0.93). For the ACQ-SF-R factor scores, the means were 1.86 (SD = 1.13) for Emotionality, 2.74 (SD = 1.59) for Purposefulness, and 1.56 (SD = 0.86) for Compulsivity (Table 1).

Significant differences were also found between groups in the PACS [$F(4, 90.464) = 83.778$, $p < 0.001$, $\omega^2 = 0.373$] and the ACQ-SF-R [$F(4, 95.423) = 91.151$, $p < 0.001$, $\omega^2 = 0.302$] total scores (Fig. 2). *Post-hoc* analysis revealed significant differences in all group comparisons for the PACS score, showing a linear progression from Abstainers to Dependent Drinkers. Specifically, Dependent Drinkers displayed the highest scores compared to Binge Drinkers ($p = 0.004$), Hazardous Drinkers ($p < 0.001$), Moderate Drinkers ($p < 0.001$), and Abstainers ($p < 0.001$). Binge Drinkers scored higher than

Hazardous Drinkers ($p = 0.004$), Moderate Drinkers ($p < 0.001$) and Abstainers ($p < 0.001$). Hazardous Drinkers had higher scores than Moderate Drinkers ($p < 0.001$), and Abstainers ($p < 0.001$). Lastly, Moderate Drinkers scored higher than Abstainers ($p < 0.001$). For the ACQ-SF-R score, *post-hoc* tests revealed that Dependent Drinkers had the highest scores, with significant differences compared to Hazardous Drinkers ($p = 0.030$), Moderate Drinkers ($p = 0.002$), and Abstainers ($p < 0.001$). Binge Drinkers had higher scores compared to Hazardous Drinkers ($p < 0.001$), Moderate Drinkers ($p < 0.001$), and Abstainers ($p < 0.001$). Hazardous Drinkers scored higher than Moderate Drinkers ($p < 0.001$) and Abstainers($p < 0.001$). Finally, Moderate Drinkers had higher scores compared to Abstainers ($p < 0.001$). Further detailed results for the group differences of each ACQ-SF-R factor score are available in the Supplemental Material.

### Sex-related differences

A significant association was found between sex and drinking group distribution [$\chi^2(4, N = 1,732) = 57.82$, $p < 0.001$]. Among female students, the largest proportion was classified as Moderate Drinkers (36.4%), followed by Hazardous Drinkers (27.8%), Abstainers (18.6%), Binge Drinkers (16.1%), and Dependent Drinkers (1.0%). Among male students, Moderate Drinkers (32.1%) and Binge Drinkers (31.0%) were the largest proportions, followed by Hazardous Drinkers (21.6%), Abstainers (13.2%), and Dependent Drinkers (2.2%).

**Drinking characteristics.** Among alcohol consumers, male students showed a significantly earlier age of drinking onset [$U = 183,785.50$, $p < 0.001$], greater percentages of drunkenness [$U = 193,831.00$, $p < 0.001$] and a higher number of standard weekly drinks [$U = 172,610.00$, $p < 0.001$] in comparison with female students. Similarly, AUDIT total scores were higher among male students than female students [$U = 170,845.00$, $p < 0.001$].

**Drinking patterns.** A greater proportion of males (68.1%) than of females (51.3%) reported having previously consumed 5 or more alcoholic drinks on a single occasion. Male students also consumed alcohol at a faster rate than female students [$U = 261,812.50$, $p < 0.001$]. While males most frequently reported consuming 2 drinks per hour (28.4%), females most commonly reported consuming one drink in 2 hours (27.9%).

**Alcoholic beverage consumption.** As for the most consumed type of alcoholic beverage, similar patterns were observed among male (beer: 61.4%; shots: 46.6%) and female students (beer: 35.9%; shots: 41.8%). Male students reported significantly higher consumption of wine [$U = 206,015.00$, $p < 0.001$], beer [$U = 168,075.00$, $p < 0.001$], and cocktails [$U = 217,170.50$, $p = 0.014$] when compared to female students. In contrast, female students reported higher consumption of sangria [$U = 207,495.00$, $p < 0.001$] than their male peers.

**Substance use.** Polydrug use was significantly different between sexes [$U = 299,796.00$, $p < 0.001$], with 32.6% of male and 24.7% of female students reporting the use of at least two combined substances. Similar patterns were observed for males (tobacco: 25.9%;

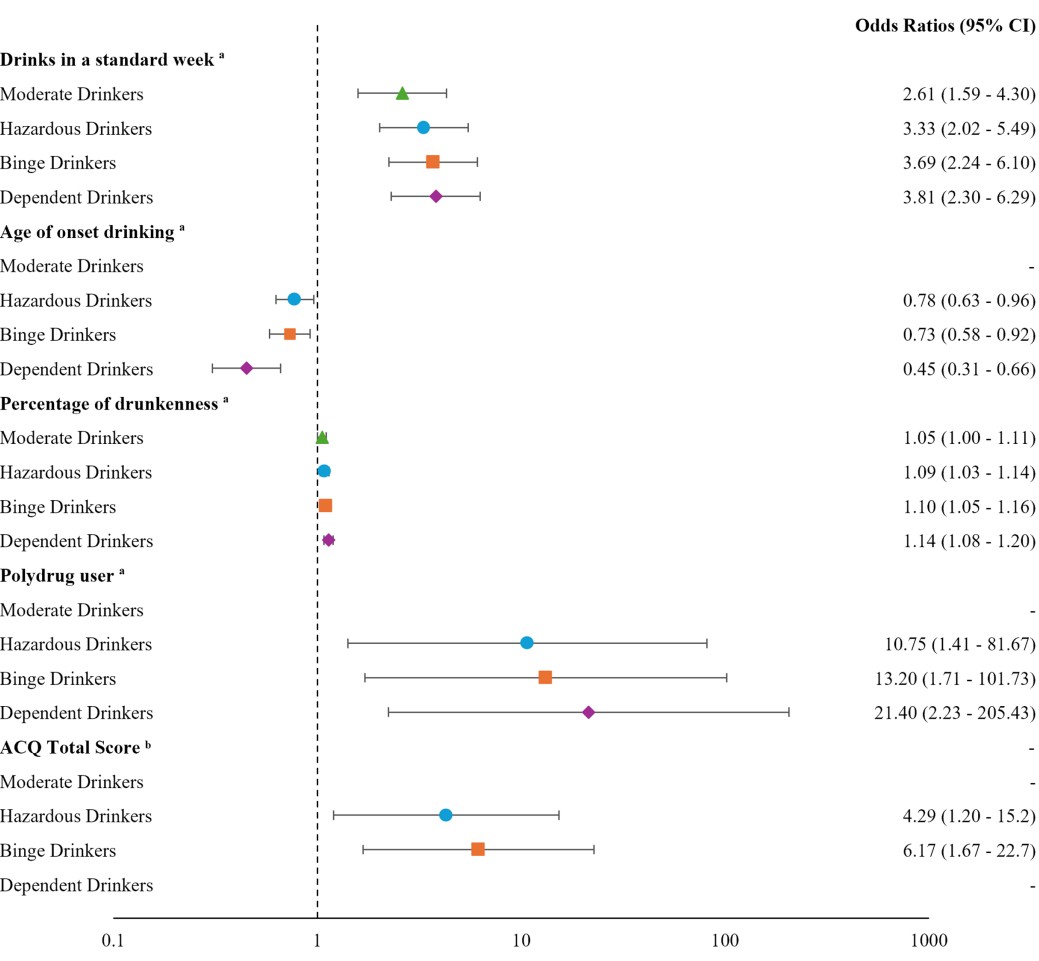

Odds Ratios (95% CI)

**Drinks in a standard week [a]**
Moderate Drinkers — 2.61 (1.59 - 4.30)
Hazardous Drinkers — 3.33 (2.02 - 5.49)
Binge Drinkers — 3.69 (2.24 - 6.10)
Dependent Drinkers — 3.81 (2.30 - 6.29)
**Age of onset drinking [a]**
Moderate Drinkers — -
Hazardous Drinkers — 0.78 (0.63 - 0.96)
Binge Drinkers — 0.73 (0.58 - 0.92)
Dependent Drinkers — 0.45 (0.31 - 0.66)
**Percentage of drunkenness [a]**
Moderate Drinkers — 1.05 (1.00 - 1.11)
Hazardous Drinkers — 1.09 (1.03 - 1.14)
Binge Drinkers — 1.10 (1.05 - 1.16)
Dependent Drinkers — 1.14 (1.08 - 1.20)
**Polydrug user [a]**
Moderate Drinkers — -
Hazardous Drinkers — 10.75 (1.41 - 81.67)
Binge Drinkers — 13.20 (1.71 - 101.73)
Dependent Drinkers — 21.40 (2.23 - 205.43)
**ACQ Total Score [b]**
Moderate Drinkers — -
Hazardous Drinkers — 4.29 (1.20 - 15.2)
Binge Drinkers — 6.17 (1.67 - 22.7)
Dependent Drinkers — -

▲ Moderate Drinkers   ● Hazardous Drinkers   ■ Binge Drinkers   ◆ Dependent Drinkers

*Reference Category: Abstainers
[a] Nunber of subjects measures included: 1,390
[b] Number of subjects measures included: 508

**Figure 3 Odds ratios (95% CI) for variables associated with drinking groups.** Forest plot showing the ORs and corresponding 95% CI (indicated by right-side labels) for variables significantly associated with the different drinking groups (left-side labels). The ORs are represented by the markers, and the 95% CI by the error bars. The vertical dashed line represents an OR of 1 (no effect). ORs below 1 indicate protective factors, where odds decrease as the predictor increases, while ORs above 1 indicate risk factors, *i.e.*, odds increase as the predictor increases. *Reference category: Abstainers; [a]Number of subjects measures included: 1,390; [b]Number of subjects measures included: 508.

cannabis: 18.1%) and females (tobacco: 21.1%; cannabis: 8.5%), with male students having significantly higher consumption of tobacco [$U = 310,034.00$, $p = 0.026$] and cannabis [$U = 294,280.00$, $p < 0.001$].

**Craving levels.** Significant differences were found between male and female students, with males reporting higher craving levels across all measures: PACS total score [$U = 39,558.50$,

$p < 0.001$], ACQ-SF-R total score [U = 59,096.50, $p < 0.001$], and Emotionality [U = 59,338.50, $p < 0.001$], Purposefulness [U = 65,379.50, $p = 0.004$], and Compulsivity factor scores [U = 63,240.00, $p < 0.001$].

### Predictors for drinking groups

A multinomial logistic regression analysis was conducted to examine the association between drinking characteristics and the drinking groups. ORs and 95% confidence intervals (95% CI) were obtained to assess the likelihood of group membership relative to the Abstainers group, which served as the reference category (Fig. 3).

Results showed that the number of drinks in a standard week, age of onset of drinking, percentage of drunkenness, polydrug use, and the ACQ-SF-R total score were significant predictors. A higher number of drinks consumed in a standard week was associated with an increased risk of becoming a Moderate Drinker (OR = 2.61), Hazardous Drinker (OR = 3.33), Binge Drinker (OR = 3.69), and Dependent Drinker (OR = 3.81). A higher percentage of alcohol consumption resulting in drunkenness was also a risk factor for being classified as a Moderate Drinker (OR = 1.05), Hazardous Drinker (OR = 1.08), Binge Drinker (OR = 1.10), and Dependent Drinker (OR = 1.14). Higher craving scores on the ACQ-SF-R were found to be a risk for becoming a Hazardous Drinker (OR = 4.29) and Binge Drinker (OR = 6.17). Polydrug use accounts for the most significant risk factor for becoming a Hazardous Drinker (OR = 10.75), Binge Drinker (OR = 13.20), and Dependent Drinker (OR = 21.40). Lastly, later ages of drinking onset were a protective factor against becoming a Hazardous Drinker (OR = 0.78), Binge Drinker (OR = 0.73), and Dependent Drinker (OR = 0.45).

### DISCUSSION

The current study aimed to provide a comprehensive and detailed description of alcohol consumption patterns among Portuguese university students, with a particular focus on adolescents and young adults. It explored predominant drinking patterns, associated characteristics and predictors, levels of craving, and concurrent substance use. Our findings align with previous research, revealing high levels of alcohol use in this population (83% were current drinkers), with similar patterns observed across genders. Nearly 47% of students revealed harmful drinking patterns, such as hazardous or binge drinking, with an additional 1.5% of students exhibiting symptoms of alcohol dependence. Furthermore, significant differences in drinking behaviours were identified across five groups: Abstainers, Moderate Drinkers, Hazardous Drinkers, Binge Drinkers, and Dependent Drinkers. The results highlighted a clear progression in drinking severity, with Dependent Drinkers reporting the highest levels overall. Significant predictors of drinking behaviours included polydrug use, higher weekly alcohol consumption, earlier drinking onset, and higher levels of alcohol craving. Additionally, variations in drinking characteristics, including sex differences, were observed.

The high rates of alcohol use observed in the present sample were consistent with those reported in other Portuguese university populations–*e.g.*, 86.8% in the study

by *Gonçalves & Carvalho (2017)*. Notably, Abstainers accounted for 16.8% of the sample, a relatively high prevalence compared to previous studies (*Pimentel, Mata & Anes, 2013*; *Alcântara da Silva et al., 2015*), thus suggesting a potential increase in abstinence rates in Portugal, as highlighted by *Intervention Service for Addictive Behaviors and Dependencies (2023)*. This trend may reflect a cultural shift among youth, who increasingly question the social and personal benefits of alcohol use and may be devaluing its social status (*Kraus et al., 2020*). Alternatively, part of this group may have been first-year students, who had not yet initiated consumption. Most students were Moderate Drinkers (35.1%), reflecting the predominance of lower-risk drinking patterns, consistent with the consumption profile typically observed among European university students (*Gonçalves & Carvalho, 2017*; *Cooke et al., 2019*; *Ay et al., 2025*). Among the riskier drinking patterns, 25.8% of students were classified as Hazardous Drinkers, and 20.8% were Binge Drinkers. These findings align with global estimates of binge drinking prevalence among young people in Western countries, ranging from 20% to 40% (*Archie, Kazemi & Akhtar-Danesh, 2012*; *Ladner et al., 2014*; *Lukács et al., 2021*; *Observatorio Español de las Drogas y las Adicciones, 2024*; *Patrick et al., 2024*; *World Health Organization, 2024*). Lastly, the smallest yet most concerning group included Dependent Drinkers, with a prevalence of 1.5%. This rate exceeds previous estimates for this age group in Portugal (*Intervention Service for Addictive Behaviors and Dependencies, 2023*), but is comparable to broader age ranges, such as the 1.4% reported by *Moreira et al. (2020)* and the 1.2% reported by *Moutinho, de Oliveira Cruz Mendes & Lopes (2018)*.

Several significant predictors of group membership were identified. Polydrug use emerged as the strongest predictor, particularly for Hazardous Drinkers, Binge Drinkers, and Dependent Drinkers. This type of substance use had a high prevalence overall (27.3%), increasing among Binge Drinkers and Dependent Drinkers. The significant association highlights the use of other substances while drinking as a key risk factor for developing problematic drinking behaviours. Accordingly, research underscores this pattern as common among heavy drinkers (*O'Grady et al., 2008*) and individuals with alcohol dependence (*Moss et al., 2015*). In line with this, tobacco (22.7%) and cannabis (11.5%) were the most consumed substances across all groups, particularly among Binge Drinkers (44.6% and 27.0%, respectively) and Dependent Drinkers (73.1% and 34.6%, respectively), with studies showing that their use is associated with increased alcohol consumption (*Wicki, Kuntsche & Gmel, 2010*; *Mostardinha & Pereira, 2020*; *Lukács et al., 2021*; *Cerqueira et al., 2022*).

A strong link was identified between the number of alcoholic drinks consumed in a standard week and the likelihood of belonging to more severe drinking groups. Given that group classification was inherently linked to alcohol consumption (with AUDIT items assessing frequency and quantity), these results are best interpreted as supporting internal consistency and grouping validity, rather than as evidence of independent predictive value. However, no significant differences were observed in the total consumption levels between Binge Drinkers and Dependent Drinkers. This may reflect the distinct drinking patterns of these groups: Binge Drinkers tend to concentrate their alcohol intake into a single

occasion, while Dependent Drinkers distribute their consumption more evenly throughout the week (*Courtney & Polich, 2009*). Alternatively, some studies have proposed the concept of high-intensity binge drinking–defined as consuming two or more times the standard binge threshold–as being present among university students (*Patrick & Azar, 2018*). Although this study does not directly assess this subtype, the similar consumption levels observed between Binge Drinkers and Dependent Drinkers may reflect overlapping psychological risk profiles and drinking motives (*Patrick & Terry-McElrath, 2021*) that approach those more typically associated with alcohol dependence, further emphasising the progressive severity between the groups. Notably, these findings highlight the importance of policies targeting overall alcohol reduction at these ages, especially considering the widespread binge drinking behaviour observed among young Portuguese university students (with a prevalence exceeding 20% of the student population).

Similarly, craving, as measured by ACQ-SF-R scores, also emerged as a significant predictor of group membership, with Dependent Drinkers consistently reporting the highest levels of craving. Interestingly, this measure did not emerge as a significant risk factor for becoming a Dependent Drinker, although this might be attributed to the smaller number of subjects included in that specific analysis. Overall, our results did not indicate clinically significant craving, with mean scores pointing to the low end on both ACQ-SF-R (M = 1.99) and PACS (M = 2.01). However, more severe drinking patterns were associated with higher craving levels, particularly in relation to goal-directed motivations, supporting the idea that enhancement and social motives are prevalent among this population (*Kuntsche et al., 2014*; *Lannoy et al., 2017*). Further, no significant differences were found between Dependent Drinkers and Binge Drinkers, reinforcing the presence of overlapping characteristics between the two groups. These results support the idea that binge drinking and alcohol dependence may share common neurobiological and cognitive mechanisms driving alcohol consumption, particularly those closely linked to craving, such as heightened alcohol-cue reactivity, increased attentional bias towards alcohol-related stimuli, and inhibitory control abnormalities (*Østby et al., 2009*; *López-Caneda et al., 2017*; *Lannoy et al., 2019*; *Lees et al., 2019*). This idea of *continuum* between binge drinking and AUD emphasises the potential progression in drinking severity, where individuals engaging in binge drinking behaviours may be at an increased risk of transitioning toward alcohol dependence (*Enoch, 2006*; *Hingson, Zha & White, 2017*; *Addolorato et al., 2018*; *Tavolacci et al., 2019*).

Early onset of drinking has also been widely recognised as a risk factor for developing harmful drinking patterns later in life (*Grant, Stinson & Harford, 2001*; *Hingson, Heeren & Winter, 2006*; *Morean et al., 2014*), a finding that is further supported by our results. On average, students reported initiating alcohol consumption at age 16, with Dependent Drinkers showing the earliest onset, at 14.8 years. The linear progression from Abstainers to Dependent Drinkers underscores the cumulative impact of early alcohol exposure on future drinking patterns and the development of AUD (*Grant et al., 2015*; *Hingson, Heeren & Winter, 2006*; *National Institute on Alcohol Abuse and Alcoholism, 2024a*), highlighting

the importance of delaying the onset of alcohol consumption in public health policies aimed at reducing the risk of alcohol misuse and its associated harms (*Hingson & Zha, 2009*; *Stigler, Neusel & Perry, 2011*).

The frequency with which students reported getting drunk when drinking was also identified as a significant risk factor for more harmful consumption patterns. While this measure is subject to inconsistencies and reliability issues due to differences in accuracy and varying definitions of drunkenness (*Greenfield & Kerr, 2008*; *Müller et al., 2011*; *Kilian et al., 2020*), it continues to offer valuable information regarding consumption patterns (*Maurage et al., 2020*; *André et al., 2023*). Overall, students indicated that they get drunk nearly one-third (28%) of the time they drink alcohol. In the last year, the prevalence of drunkenness among Portuguese consumers in this age group was estimated at 18.4% (*Intervention Service for Addictive Behaviors and Dependencies, 2023*), and studies targeting university students revealed higher rates, such as 51.8% (*Ferreira Alves, Precioso & Becoña, 2021*) and 43.3% (*Pombo & Sampaio, 2010*). Among university students, the most common reasons for alcohol consumption are linked to enhancement and social motives, with some emphasising drinking to get drunk (*Kuntsche et al., 2005*; *Cunha, 2014*) or achieve a tipsy state (*Beccaria, Petrilli & Rolando, 2015*).

Regarding the types of beverages consumed, beer emerged as the most commonly consumed alcoholic drink across all groups, followed by shots. This finding goes in line with previous studies, where beer stands out as the preferred beverage among young drinkers (*Pimentel, Mata & Anes, 2013*; *Araújo & Medeiros, 2020*; *Beccaria & Pretto, 2021*), likely due to its lower cost and strong presence in festivities among this age group (*Cunha, 2014*; *Rodrigues et al., 2014*). The analyses revealed that the types of alcoholic beverages consumed were highly interrelated and did not uniquely differentiate behavioural outcomes or group membership. In practice, students often consume multiple types of alcohol interchangeably (*Smart & Walsh, 1995*). These results support the notion that alcohol consumption patterns among university students are better understood through broader indicators of quantity and risk level, rather than beverage-specific effects (*Dey et al., 2014*).

Consistent with the literature, significant sex differences were also found, with male students exhibiting higher levels of alcohol consumption than their female peers, leading to a greater risk of alcohol dependence (*Wicki, Kuntsche & Gmel, 2010*; *Grant et al., 2015*; *Carvalho et al., 2019*). Males also reported higher craving levels than females, further emphasising the importance of gender in understanding alcohol use patterns (*Erol & Karpyak, 2015*). Nonetheless, the narrowing of prevalences among sexes, reflected in the more equal distribution in the more severe drinking groups, followed an important trend among adolescents and youth observed in several studies (*Davoren et al., 2016*; *Slade et al., 2016*; *Edkins, Edgerton & Roberts, 2017*). This approximation of alcohol use is associated with an increase in consumption by females, while male consumption decreases or remains the same (*Wilsnack et al., 2018*; *White, 2020*).

Taken together, the present findings underscore the importance of early intervention to address risky and harmful drinking behaviours, particularly among students who begin

drinking at a young age and exhibit signs of problematic drinking. Prevention programmes should focus on delaying the onset of alcohol consumption, address craving-related factors, and call attention to the risks of concurrent drug use and its potential to exacerbate alcohol-related harms, particularly those more prevalent at these ages. Furthermore, promoting alcohol-free academic festivities could be crucial in developing healthier social norms within university environments. Despite Portugal being recognised with some level of safety regarding drinking behaviours and deemed successful in addressing drinking-related problems (*Pentz et al., 2023*), further efforts are needed to address the emerging challenges related to alcohol use in adolescents and young adults. University settings, in particular, should consider integrating early screening protocols and psychoeducation campaigns not only targeting binge episodes but also focusing on the psychological mechanisms underpinning problematic drinking.

Future research should explore the role of other psychological factors, such as personality traits and mental health conditions, in shaping drinking behaviours. Longitudinal studies that track changes in drinking patterns over time will also be invaluable for understanding this trajectory of alcohol use. For instance, *Moure-Rodriguez et al. (2018)* observed a marked decrease in risky and binge drinking patterns after the age of 24, likely due to the completion of studies and the transition into the work environment. Similarly, *Busto Miramontes et al. (2020)* found that while the consumption of alcohol, tobacco, and cannabis declines over time among university students, the use of medication increases, suggesting the emergence of a different pattern of polydrug use. Such findings highlight the importance of analysing how life transitions and developmental stages influence alcohol consumption behaviours over time.

While this study provides valuable insights, several limitations should be considered. First, not only does the cross-sectional design prevent the generalisation of the results for all university students, but the inclusion of only students from the University of Minho also limits these findings to northern Portugal. However, comparable results have been reported by other studies from the northern region (*Moreira et al., 2020*; *Ferreira Alves, Precioso & Becoña, 2021*), the central region (*Santos, Veiga & Pereira, 2009*; *Coelho, 2010*), and Lisbon (*Moutinho, de Oliveira Cruz Mendes & Lopes, 2018*), suggesting some consistency across different regions. Furthermore, the reliance on self-report measures may introduce bias, particularly in reporting alcohol and drug use (*Davis, Thake & Vilhena, 2010*; *Latkin et al., 2017*). Additionally, missing responses for some variables resulted in smaller sample sizes for certain analyses, which limited the statistical power to detect significant effects (increased Type II error) and may have increased the likelihood of unstable estimates and false-positive findings (increased Type I error). Therefore, the results of these analyses should be interpreted with caution and considered exploratory until replicated in larger samples. Moreover, some descriptive findings (*e.g.*, speed of consumption) overlap with the criteria used for group classification. As such, these served as indicators of internal consistency and were reported without interpretive weight to avoid overstating their contribution. Some variables that would provide a more

comprehensive understanding of youth consumption patterns, such as socio-economic status, geographic location, living situation, academic level, parental education, and parental alcohol/substance use, were not collected. This omission may limit the scope of the present study, as these factors are known to significantly influence consumption behaviours in youth (*Borsari, Murphy & Barnett, 2007*; *Wicki, Kuntsche & Gmel, 2010*; *Kuntsche et al., 2017*). The presence of psychological conditions, particularly mood disorders, was also not assessed, despite their high prevalence in university students (*Auerbach et al., 2018*). As early-onset mood disorders have been associated with an earlier initiation of alcohol use among youth (*Crum et al., 2008*; *Birrell et al., 2016*), these factors should be included in future studies as they may be important predictors for risky drinking behaviours.

Another limitation lies in the lack of consensus regarding the definition of alcohol consumption and the categorisation of drinking patterns, not only in the assessment instruments employed but also in their cut-off points, which may vary significantly across countries and cultural contexts (*Knibbe et al., 2006*; *Nadkarni et al., 2019*), complicating the interpretation of the results.

Nonetheless, while prior studies have mapped alcohol consumption among Portuguese youth, this study adds value in several critical ways. First, it focuses on a specific developmental period (17–24 years), a stage during which drinking patterns are likely to consolidate, thereby avoiding the interpretative limitations associated with broader age ranges–*e.g.*, 18–54 years (*Ferreira Alves, Precioso & Becoña, 2021*), 18–63 years (*Moreira et al., 2020*). Second, our study combines a robust sample of nearly 1,800 university students, and although some studies employed larger samples (*Alcântara da Silva et al., 2015*), many lacked the multidimensional assessment used here (*e.g.*, craving, polydrug use, age of drinking onset, and binge criteria beyond AUDIT). Lastly, this study provides updated epidemiological evidence with direct relevance for prevention policies: delaying the onset of alcohol use, screening for polydrug behaviours, and monitoring craving-related factors may reduce the transition to more severe forms of alcohol misuse.

## CONCLUSIONS

This study provides valuable insights into the alcohol consumption patterns among Portuguese university students, spanning from low-risk to high-risk drinkers. Our findings highlight elevated levels of alcohol consumption, with particularly concerning trends in binge drinking and alcohol dependence among university students. Significant predictors of severe drinking behaviours, such as polydrug use and early onset of alcohol consumption, were identified, emphasising critical areas for targeted intervention. While Portugal has made progress in addressing alcohol-related harms, our results underscore the need for early screening protocols and prevention programmes focused on specific risk factors, including delaying the onset of alcohol use and/or reducing drug use. Such initiatives prove essential to mitigate the risks associated with alcohol use and foster healthier behaviours among university students.

## ACKNOWLEDGEMENTS

We thank the university students who have voluntarily made their time available to complete the questionnaires.

### Funding

This study was conducted at the Psychology Research Center (PSI/01662), School of Psychology, University of Minho, supported by the Foundation for Science and Technology (FCT) through the Portuguese State Budget (Ref.: UIDB/PSI/01662/2020). This study was also supported by the projects PTDC/PSI-ESP/1243/2021 and 2023.14679. PEX. Eduardo López Caneda and Alberto Crego were supported by the FCT and the Portuguese Ministry of Science, Technology and Higher Education, within the scope of the Individual Call to Scientific Employment Stimulus (CEECIND/07751/2022), and the Transitory Disposition of the Decree No. 57/2016, of 29 August, amended by Law No. 57/2017 of 19 July, respectively. Rui Rodrigues (2021.05276.BD) was supported by the FCT, MCTES, and European Union through the European Social Fund. The funders had no role in study design, data collection and analysis, decision to publish, or preparation of the manuscript.

### Grant Disclosures

The following grant information was disclosed by the authors:
Foundation for Science and Technology (FCT) through the Portuguese state Budget: UIDB/PSI/01662/2020, PTDC/PSI-ESP/1243/2021, 2023.14679.PEX.
Rui Rodrigues was supported by FCT: 2021.05276.BD.

### Competing Interests

The authors declare that they have no competing interests.

### Author Contributions

- Lucas Saldanha analyzed the data, prepared figures and/or tables, authored or reviewed drafts of the article, and approved the final draft.
- Alberto Crego conceived and designed the experiments, authored or reviewed drafts of the article, and approved the final draft.
- Natália Almeida-Antunes performed the experiments, prepared figures and/or tables, authored or reviewed drafts of the article, and approved the final draft.
- Rui Rodrigues performed the experiments, authored or reviewed drafts of the article, and approved the final draft.
- Adriana Sampaio conceived and designed the experiments, authored or reviewed drafts of the article, and approved the final draft.
- Eduardo López-Caneda conceived and designed the experiments, performed the experiments, analyzed the data, prepared figures and/or tables, authored or reviewed drafts of the article, and approved the final draft.

# PeerJ

## Human Ethics

The following information was supplied relating to ethical approvals (*i.e.*, approving body and any reference numbers):

The Institutional Ethics Committee for Social Sciences and Humanities of the University of Minho granted ethical approval (CECSH 078/2018) for this research.

## Data Availability

The data is available at GitHub and Zenodo:

- https://github.com/lsaldanhar/Abstainers-to-Dependent-Drinkers-Raw-Data.git.

- LS. (2025). lsaldanhar/Abstainers-to-Dependent-Drinkers-Raw-Data: Initial Data Release (v1.0). Zenodo. https://doi.org/10.5281/zenodo.15838653.

## Supplemental Information

Supplemental information for this article can be found online at http://dx.doi.org/10.7717/peerj.20026#supplemental-information.

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
