# Peer review of "From Abstainers to Dependent Drinkers: alcohol consumption patterns and risk factors among Portuguese university students"

_PeerJ, doi:10.7717/peerj.20026_

## Round 0.1 · original submission · Major Revisions

I have read the feedback from each of the reviewers and recommend that you revise your manuscript in line with their comments and the PeerJ editorial criteria: https://peerj.com/about/editorial-criteria/.

Reviewer 1 ·

Basic reporting

Major revisions:
1. As social norms play an important role in drinking behaviors among university students, I would recommend discuss it in more detail (lines 105-108). Indeed, the literature has shown that college students frequently overestimate descriptive and injunctive alcohol consumption norms on campus (e.g., Borsari & Carey, 2001). In turn, these misperceptions can make excessive alcohol use appear more common and socially acceptable or lead students to feel pressured to conform to these inflated norms, increasing their alcohol consumption (e.g., Borsari & Carey, 2001; Crawford & Novak, 2007; França et al., 2010). A more detailed discussion of the specific influence of these norms on the student population would enhance the study’s contribution, highlighting the added value of focusing on a student-specific sample compared to prior broader studies. Conversely, I’m not sure the “Mediterranean drinking pattern, where alcohol is typically consumed during meals” apply to students.
2. Similarly, developing the specific effect of alcohol during a developmental period (line 150-151) would strengthen the relevance of this study focusing specifically on youths. Authors should define for the reader what a developmental period is and explain why alcohol consumption during this period is particularly noxious (e.g., Bava & Tapert, 2010; Crews et al., 2007; Østby et al., 2009).

Minor revisions:
1. I was not able to find the raw data. Make sure the correct file has been imported.
2. Abstract: The classifications in alcohol consumption groups in described in the “method” section while it is actually the result of the present paper (lines 52-54).
3. Abstract: “A progressive increase in the severity of alcohol consumption characteristics was observed across the groups, with Dependent Drinkers reporting the highest levels overall”. Levels of what?
4. Abstract: “while Moderate Drinkers exhibited the least risky patterns, including a later age of onset of drinking”. Authors should specify “among drinkers” because the reader might wonder why the Abstainers are not the least risky pattern.
5. Abstracts: “Programs” and not “programmes” (line 71)
6. Introduction: “characterised by” instead of “characterised as” (line 112)
7. Results: Figure 1A is not mentioned in the manuscript. However, this figure is not particularly informative, as the categories themselves are defined based on the AUDIT score, making differences between groups on this measure tautological.
8. Results: I recommend removing the significance bars from Figure 2, as they clutter the visual presentation and offer limited added value if all groups differ significantly. Additionally, some of the significant differences reported in the text are missing from the figure: the difference between Hazardous Drinkers and Abstainers on the PACS, and between Dependent Drinkers and Abstainers on the ACQ-SF-R.
9. Discussion: “abstinence” and not “abstention” (line 409)

Experimental design

Major revisions:
3. Authors provide a lot of previous data about alcohol consumption – and binge drinking – worldwide but also in Portugal specifically (lines 116-122; 130-137; 522-525). While epidemiological data are inherently valuable, this article contributes little new information, given that similar studies on this topic have already been conducted and the present work does not offer significant additional insights. So while the methodological rationale is acceptable (i.e., different age range, post-covid, bigger sample size), the manuscript currently lacks a strong positioning in the Introduction and Discussion regarding the broader significance of the findings. Strengthening these sections would help clarify the contribution of the present study beyond descriptive replication. The study doesn’t frame any specific hypotheses about the potential changes – or not – in alcohol consumption and doesn’t explain what implications the new data would have. I strongly recommend clarifying the contribution of the present study in the Introduction and Discussion, and discuss the practical implications of the findings in more detail. The authors argue that previous studies used small sample size which is not the case of all of them (e.g., Alcântara da Silva et al., 2015), which should be discussed honestly.
4. The authors argue that binge drinking is poorly defined in previous studies with different classification criteria (lines 122-123), which highlight the need for a clear, consensual definition of binge drinking. However, the rationale for selecting the three criteria used to assess binge drinking is not clearly explained. It would be important to justify why alternative measures (e.g., binge drinking score by Townshend & Duka (2002, 2005)) were not considered, especially given that the chosen criteria may be insufficient to fully characterize binge drinking in accordance with recent guidelines (e.g., Maurage et al., 2020).
In the same vein, authors highlight the lack of consensus regarding the definition and categorization of drinking patterns (lines 534-538) but do not specify or justify which norms they followed in the present study (e.g., “AUDIT score of ≤ 4 for men or ≤ 3 for women were classified as Moderate Drinkers”).

Minor revisions:
1. Measures: Authors do not specify how other substance consumption was evaluated (likert scale?). They talk about frequency in the Measures section but in the Discussion: “tobacco (22.7%) and cannabis (11.5%) were the most consumed substances across all groups, particularly among Binge Drinkers (44.6% and 27.0%, respectively) and Dependent Drinkers (73.1% and 34.6%, respectively)”, which suggests participants were either consumer or not. A clear description of the substance consumption assessment is needed, specifically because smoking one cigarette occasionally is different from presenting a tobacco use disorder.
2. Discussion: “As expected, most students were Moderate Drinkers”. Authors do not make any hypotheses in the paper and do not justify why this result was expected.

Validity of the findings

Major revisions:
1. The sex distribution paragraph (lines 261-267) is totally misleading as authors argue that most alcohol consumption groups are predominantly female which totally normal as the whole sample is composed of almost 70% of women. Authors should add in this paragraph that the sample is 67.6% female, 31.6% male. I recommend that the authors reword this paragraph to present the proportion of females within each category relative to the total number of women in the sample (e.g., '30% of women in the sample are binge drinkers' rather than 'the binge drinker category is 70% female'), which would provide a clearer interpretation of gender-related patterns.
2. Authors do not justify the choice of the predictors included in this study that seem restrictive. “Results showed that the number of drinks in a standard week, age of onset of drinking, percentage of drunkenness, polydrug use, and the ACQ-SF-R total score were significant predictors.” (lines 377-379, lines 402-403). While some of these predictors are interesting (e.g., craving), others seem redundant (obviously the weekly alcohol consumption predicts the alcohol consumption group) and I don’t see the relevance of such result. Authors should clarify the usefulness of the predictors chosen and justify why they did not include other relevant predictors (e.g., familial history of alcohol dependence, socio-economic status, psychological distress, perceived norms, etc).

Minor revisions:
1. Results: “post-hoc analysis revealing significant differences between all group comparisons, except between Binge Drinkers and Dependent Drinkers”. This an interesting result that should be further discussed, highlighting the similarities and differences between these two consumption patterns and what make them unique.
2. Results: Several of the reported results appear tautological and offer limited interpretative value (e.g., lines 291–296). For example, noting that “Binge Drinkers reported the highest frequencies for drinking at speeds of 2 or 3 drinks per hour” is expected, given that this drinking speed was part of the criteria used to define binge drinking in the study.
3. Results: The authors should elaborate on the relevance of the “alcoholic beverages consumption” sections (lines 320-333, 481-487). For instance, does the type of alcohol consumed have any association with other key variables or behavioral outcomes?
4. Results: “Polydrug use patterns also varied across groups, with Binge Drinkers showing the highest prevalence (41.2%)”. The figure S3 seems to contradict this result as we can see that polydrug users account for 54% of binge drinkers and 73.1% of dependent drinkers. “The use of other drugs – such as hallucinogens, amphetamines, cocaine, ecstasy, and heroin – was relatively low across groups, except for heroin, which had a notably higher rate of use among Dependent Drinkers (11.5%)”. Here again, the figure S3 shows the same pattern in dependent drinkers for amphetamines (11.5%), cocaine (15.4%) and ecstasy (15.4%). The results are thus misleading as they suggest this effect is only present for heroin.
5. Results: Authors report the mean craving scores without further discussion. Is a score of 2 high? Low? How should these results be interpreted?
6. Results: Higher craving scores on the ACQ-SF-R being a predictor for hazardous drinking and binge drinking but not for alcohol dependence is a surprising result as craving is one of the DSM-V criteria for severe alcohol use disorder. This should be discussed.
7. Discussion: “Abstainers accounted for 16.8% of the sample, a relatively high prevalence compared to previous studies (Pimentel, Mata & Anes, 2013; Alcântara da Silva et al., 2015), thus suggesting a potential increase in abstention rates in Portugal, as highlighted by SICAD (2023)”. The authors are encouraged to explore possible explanatory factors underlying the increased rate of abstinence reported in the study.

Additional comments

Thank you for the opportunity to review this manuscript. The study addresses an important public health topic and brings useful data specifically focusing on Portuguese university students, with a notably large sample size compared to prior studies. Overall, I recommend a major revision to better articulate the study’s relevance and implications.

Reviewer 2 ·

Basic reporting

The manuscript meets the Basic Reporting standards. The text is written in clear, professional English. Sufficient background information and references are provided to situate the study within the existing literature. The Introduction is generally clear and well-structured; however, the explanation provided in lines 105–107 would benefit from further elaboration to better support the rationale for the study. The manuscript follows standard scientific structure, and the figures and tables are relevant, properly labeled, and of appropriate quality. The manuscript is self-contained, addressing the stated hypotheses without inappropriate fragmentation.

Experimental design

The experimental design is clearly described and well-justified.

1) The Methods section does not mention whether participants provided informed consent. Additionally, a blank copy of the informed consent form is not provided as supplementary material.

2) The use of the AUDIT questionnaire to classify participants into Moderate Drinkers, Hazardous Drinkers, Binge Drinkers, and Dependent Drinkers is appropriate in principle. However, the classification criteria used in this study—namely, AUDIT scores >5 for males and >4 for females to define 'Hazardous Drinkers'—differ from the original scoring guidelines, which recommend a threshold of ≥8 for hazardous drinking and ≥15 for dependence (Babor et al., 2001). This deviation should be clearly justified, ideally with reference to prior literature or relevant national guidelines, if applicable.

Validity of the findings

The findings are clearly presented and appear to support the authors’ conclusions.

1) To improve clarity and support interpretability, I suggest adding a brief paragraph at the end of the Results section summarizing findings related to gender differences. Alternatively, gender-related results could be incorporated as dedicated sub-paragraphs within each results subsection.

2) I also recommend expanding the Limitations section to acknowledge that mood disorders may influence alcohol consumption patterns. Relevant references might include:

- Birrell, L., Newton, N. C., Teesson, M., & Slade, T. (2016). Early onset mood disorders and first alcohol use in the general population. Journal of Affective Disorders, 200, 243–249.
- Crum, R. M., et al. (2008). Depressed mood in childhood and subsequent alcohol use through adolescence and young adulthood. Archives of General Psychiatry, 65(6), 702–712.

3) Additionally, consider summarizing the strengths of the study alongside its limitations at the end of the manuscript. For instance, in the paragraph beginning on line 519 (e.g., “While this study provides valuable insights…”). This would offer a more balanced conclusion and further emphasize the study’s contribution.

Additional comments

This is a well-conducted and clearly written study that addresses an important issue related to alcohol consumption among young adults. The methodology is rigorous, the statistical analyses are appropriate, and the findings are relevant to both clinical and research audiences. The paper would benefit from a few clarifications and additions as noted above, but these are minor and do not undermine the overall quality of the work. With these revisions, the manuscript will be a valuable contribution to the literature.

---

## Round 0.2 · Minor Revisions

After peer review, we invite you to make minor revisions based on the reviewers’ comments. If certain comments cannot be addressed, provide a clear justification. Once revised, your manuscript will be re-evaluated for final acceptance. We look forward to your resubmission.

Reviewer 1 ·

Basic reporting

“Polydrug use patterns also varied across groups, with 41.2% being Binge Drinkers”: I still don’t understand where the 41.2% come from. Does it mean that 41.2% of polydrug users are binge drinkers? In this case, the phrasing should be improved.

Experimental design

no comment

Validity of the findings

I insist on my previous comment but the inclusion of weekly alcohol consumption as a predictor of group membership raises concerns of tautology. Since the groups were defined using the AUDIT, which includes an item assessing alcohol consumption frequency and quantity, it is unsurprising that self-reported weekly consumption significantly predicts group classification. This result does not offer novel insight, but rather confirms that participants responded consistently across closely related items. While this consistency may be useful as a validity check, I do not believe it should be presented as a substantive result. As the aim was not to validate group classification, I recommend that the authors reframe this finding accordingly and clarify its interpretive value in the manuscript.
The same apply for my previous comment highlighting the tautological aspect of the result “Binge Drinkers reported the highest frequencies for drinking at speeds of 2 or 3 drinks per hour”. I appreciate the authors’ acknowledgment that some of the reported findings reflect the criteria used to define the drinking groups, and I agree that such results can support internal consistency and construct validity. However, I would encourage the authors to clearly distinguish between findings that serve a validatory function versus those that provide novel empirical insights. For instance, stating that Binge Drinkers report higher frequencies of rapid drinking confirms the consistency of classification but does not add interpretative value beyond what is already inherent in the group definitions. I recommend that such results be explicitly framed as validation checks rather than substantive findings, to avoid overstating their novelty or interpretive contribution.

Reviewer 2 ·

Basic reporting

The revised manuscript meets the Basic Reporting standards. The authors have responded adequately to the initial comments. In particular, the explanation in lines 105–107 has been expanded and now provides a more thorough rationale for the study. The manuscript is clearly written, well-structured, and appropriately referenced. Figures and tables are clear and relevant, and the manuscript is self-contained, addressing its hypotheses without fragmentation.

Experimental design

The authors have clarified the experimental procedures and addressed the points raised.
- The "Methods" section now includes a statement confirming that participants provided informed consent;
- The use of the AUDIT for participant classification has been more clearly justified. The authors now acknowledge the deviation from the original scoring criteria and provide a rationale with reference to relevant literature, which strengthens the methodological transparency of the study.

Validity of the findings

The authors have taken appropriate steps to improve clarity and contextualization of the findings.
- A concise summary of gender-related findings has been added to the Results section, facilitating interpretation;
- The "Limitations" section has been expanded to consider the potential influence of mood disorders on alcohol consumption patterns, with appropriate references included;
- Finally, a short paragraph highlighting the study's strengths has been added at the end of the manuscript, balancing the discussion of limitations and reinforcing the study’s contribution.

Additional comments

Overall, I consider the authors’ revisions to be satisfactory, and the manuscript is now suitable for publication.

---

## Round 0.3 · accepted · Accept

I have carefully reviewed the revised manuscript and confirm that the authors have satisfactorily addressed all of the reviewers’ comments. I have assessed the revision and I am satisfied with the current version. In my opinion, the manuscript is now ready for publication.